# Extinction and subsequent updating of innate fear responses to a visual looming stimulus rely on hippocampus-dependent mechanisms

Paul B. Conway[1,2], Livia Autore[1,2,3], Andrea Muñoz Zamora[1,2], Antoine Harel[1,2], Zijun Wang[1,2], Arman A. Tavallaei[1,2], Stephen M. Winston[4,5], Clara Ortega-de San Luis[1,2,6], James D. O'Leary[1,2,7], Mark A. Brimble[8], Gisella Vetere[3], Tomás J. Ryan [1,2,9,10]*

1 School of Biochemistry and Immunology, Trinity College Dublin, Dublin, Ireland, 2 Trinity College Institute for Neuroscience, Trinity College Dublin, Dublin, Ireland, 3 Cerebral Codes and Circuits Connectivity team, Brain Plasticity Unit, CNRS, ESPCI Paris, PSL Research University, Paris, France, 4 Department of Surgery, St. Jude Children's Research Hospital, Memphis, Tennessee, United States of America, 5 Graduate School of Biomedical Sciences, St. Jude Children's Research Hospital, Memphis, Tennessee, United States of America, 6 Department of Health Sciences, University of Jaén, Jaén, Spain, 7 Department of Neuroscience, Georgetown University Medical Center, Washington, District of Columbia, United States of America, 8 Department of Host-Microbe Interactions, St. Jude Children's Research Hospital, Memphis, Tennessee, United States of America, 9 Florey Institute of Neuroscience and Mental Health, Melbourne Brain Centre, University of Melbourne, Melbourne, Victoria, Australia, 10 Child & Brain Development Program, Canadian Institute for Advanced Research (CIFAR), Toronto, Ontario, Canada

* tomas.ryan@tcd.ie

## Abstract

Animals rely on innate and learned behavior to respond to their environment, but how the brain balances hardwired responses with adaptive flexibility remains unclear. Here, we demonstrate that innate looming stimulus responses in *Mus musculus* can be attenuated via repeated unreinforced presentation. This attenuation is long-lasting and generalizing, but is rapidly recovered when the stimulus is paired with an electric foot-shock. Fiber photometry recordings reveal attenuation of responses to visual looming stimuli in the SC and PAG, which do not recover following recovery of behavioral responses. Analysis of c-Fos expression uncovered a ventral CA1 (vCA1) ensemble that is active during both innate and learned looming fear responses. We report that this vCA1 engram is not necessary for innate defensive behavior but is necessary for learned fear responses. These findings reveal a novel role of the hippocampus in adapting to looming stimuli, and provide a platform for understanding the interaction of memory and instinct.

## Introduction

Innate behaviors enable an animal to respond to ethologically salient stimuli without prior experience. This is particularly advantageous for threatening stimuli, as it allows the animal to engage in appropriate behavior upon first encounter with the stimulus. However, over time the meaning of a given environmental cue can change, requiring

**Data availability statement:** All relevant data are within the paper and its Supporting information files. Code can be found: https://doi.org/10.5281/zenodo.17048095.

**Funding:** This work was funded by the Air Force Office of Scientific Research (FA9550-20-1-0316 and FA9550-24-1-0258 to T.J.R.; https://www.afrl.af.mil/AFOSR/), European Research Council (715968 to T.J.R.; https://erc.europa.eu/), Research Ireland (15/YI/3187 to T.R.; https://www.researchireland.ie/, formerly Science Foundation Ireland), Irish Research Council (GOIPG/2021/1331 to P.C. and T.J.R.; https://research.ie/, now Research Ireland), and Trinity College Dublin to T.J.R.; (https://www.tcd.ie/). The funders provide research expenses, salaries, and student stipends. The funders had no role in study design, data collection and analysis, decision to publish, or preparation of the manuscript.

**Competing interests:** The authors have declared that no competing interests exist.

**Abbreviations:** airPLS, penalized least squares; HPRA, Health Products Regulatory Authority; PAG, periaqueductal gray; PAGdm, dorsal periaqueductal gray; SC, superior colliculus; SCd, deep superior colliculus; CS, conditioned stimulus; tTA, tetracycline transactivator; vCA1, ventral CA1.

the animal to adapt their responses through learning. This process causes changes in neuronal connectivity via plasticity mechanisms, and the enduring changes which encode this novel information are known as the memory engram [1–3]. This raises an important question: how does the brain generate innate behavioral responses while still allowing for behavioral flexibility and learning?

A visual looming stimulus, such as an overhead expanding black disk which mimics an approaching aerial threat, can be used to trigger innate defensive responses of rodents such as freezing or fleeing towards shelter [4–6]. This innate behavioral response is crucial to the survival of the animal, and as such is highly conserved across species [6–12]. Interestingly, the innate behavioral response to a looming stimulus has also shown experience-dependent plasticity. For example, mice can avoid areas where looming stimuli were previously presented, demonstrating associative learning [4,13]. Furthermore, the behavioral response can be prevented by initial presentation of a very low contrast stimulus and gradually increasing contrast, showing latent inhibition of this innate response [14,15], or abolished by repeated presentation [16,17]. Taken together, these studies demonstrated a role of learning in mediating adaptive defensive behaviors, but what remains less understood is how innate and learned information may interact over longer periods of time to ensure adaptive behavior.

Recent work has shown that midbrain structures, particularly the superior colliculus (SC) and periaqueductal gray (PAG) [4,18–23] drive innate responses to looming stimuli, such as freezing and fleeing, by gating threat detection and response thresholds. The hippocampus is known to be involved in adaptive behavioral responses to learned threats [24–28] and in processing visual information including approaching stimuli [29], but it is not currently known if it may contribute to adaptive behavioral responses to innately threatening stimuli.

Here, we have designed a behavioral paradigm wherein visual looming threat responses are attenuated via repeated unreinforced presentation and rescued by pairing the stimulus with an electric foot shock. We report a loss in the responsiveness of the SC and PAG during visual looming stimulus presentation, which does not recover despite recovery of behavioral responses to the stimulus. Finally, we identify specific ensembles of cells in the ventral CA1 (vCA1) that are responsive to visual looming during naïve presentation, and later become necessary for rescued behavioral responses to the visual stimulus. This finding reveals a role for the hippocampus in adaptive responses to innately salient and meaningful stimuli.

## Results

### Visual looming threat responses can be extinguished and recovered via learning

Visual looming presentation to mice evokes a rapid defensive behavioral response, characterized by fleeing to a shelter and unconditioned freezing (Fig 1a). To investigate whether this innate behavior can be modulated by learning, we established a behavioral paradigm where threat responses to visual looming stimuli could be modified by extinction training (Fig 1b–1e). In Pavlovian fear conditioning, repeated

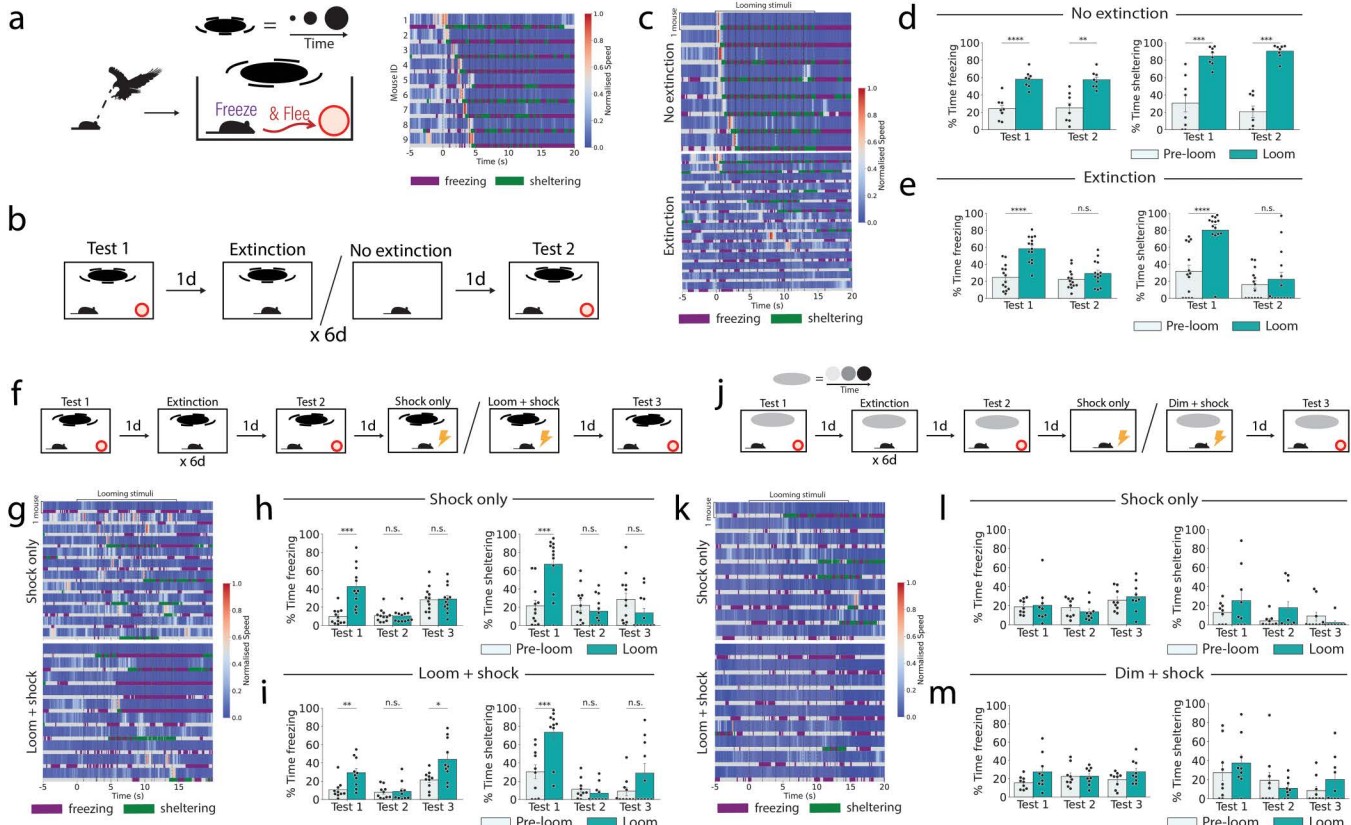

**Fig 1. Extinction and updating of innate visual looming fear responses. (a)** Behavioral paradigm to test fear responses to visual looming stimuli. The heatmap indicates speed, with each blue-red row indicating the speed of a single mouse ($n = 9$ mice). Beneath the speed, a bar indicates whether the individual mouse was detected as freezing (purple) or sheltering (green). Looming stimulus onset is indicated by the dashed black lines, with first stimulus at time $= 0$ s. Each row is the response from a single mouse. **(b)** Behavioral paradigm for extinction training. **(c)** Heatmap indicating speed for the no extinction and extinction cohorts on test 2, which each row indicting a single trial. Sheltering is indicated in green, and freezing is indicated in purple. **(d)** Freezing and sheltering behavior of the no-extinction cohort, 30 s pre- and post-stimulus onset (Student paired $t$ test; $n = 8$ mice). **(e)** As in d, but for the extinction cohort ($n = 14$ mice). **(f)** Extinction-updating training paradigm. **(g)** Heatmap indicating speed for the shock only and loom + shock cohorts on test 3, which each row indicting a single trial. Sheltering is indicated in green, and freezing is indicated in purple. **(h)** Freezing and sheltering responses of mice from the "shock only" cohort (Repeated measures ANOVA and post hoc Student paired $t$ test; $n = 12$ mice). **(i)** Freezing and sheltering responses following updating via pairing of the looming stimulus with a shock (Repeated measures ANOVA and post hoc Student paired $t$ test; $n = 10$ mice). **(j)** "Dimming" extinction-updating training paradigm. **(k)** Heatmap indicating speed on test 3 from mice undergoing the dimming-updating paradigm in either the shock only or dim + shock treatment, which each row indicting a single trial. Sheltering is indicated in green, and freezing is indicated in purple. **(l)** Freezing and sheltering responses of mice from the dimming "shock only" cohort (Repeated measures ANOVA and post hoc Student paired $t$ test; $n = 9$ mice). **(m)** Freezing and sheltering responses of mice from the dimming "dim + shock" cohort (Repeated measures ANOVA and post hoc Student paired $t$ test; $n = 9$ mice). n.s. $p \geq 0.05$, *$p < 0.05$, **$p < 0.01$, ***$p < 0.001$, ****$p < 0.0001$. Details of all statistical comparisons may be found in S1 Data. Underlying raw data may be found in S2 Data.

presentation of a conditioned stimulus (CS) in the absence of any reinforcing stimulus leads to the attenuation of the learned fear behavior [30,31]. It can be hypothesized that the visual looming response consists of an innate association between the visual stimulus and an unconditioned response, embodied by a developmentally constructed 'ingram' circuit in the brain [32].

To emulate fear extinction for an innate looming stimulus, mice were repeatedly presented with a looming stimulus over 6 days of training. In this assay, the experimental group was exposed to 6 rounds of 15 visual looming stimuli, repeated over 6 days. A control group was placed into the arena for the same length of time each day, but without presentation of

the visual looming stimulus. On test 1 (pre-extinction) mice demonstrated a significant increase in both freezing and sheltering during looming stimulus presentation relative to pre-stimulus presentation (Fig 1c–1e). On test 2 (post-extinction), there was no significant increase in freezing or sheltering following stimulus presentation. In contrast, control mice maintained freezing and sheltering responses to the visual looming stimulus. These results demonstrate that innate fear responses to the visual looming response are plastic and can be attenuated via repeated presentation of the stimulus without any concurrent reinforcing stimulus.

In fear conditioning, extinction memories tend to be context-specific and spontaneously recover after a period of 3 weeks [33,34]. To test the former for visual looming extinction, mice were placed in a novel context following training and tested as before (S1d Fig). In this case, mice retained similarly low levels of freezing and sheltering in both the familiar context and the novel context (S1e Fig). To determine whether fear responses spontaneously recover, mice were tested 3 weeks after the end of the extinction training protocol (S1f Fig). Mice maintained significantly lower levels of freezing and sheltering on test 3 relative to test 1 (S1g Fig). These results show that the extinction-like phenotype of the visual looming response is resistant to recovery in conditions which typically elicit recovery of extinguished fear-conditioned responses. This may reflect a high persistence of these adaptations, distinguishing it from extinction of classical fear conditioning.

To determine whether it is possible to recover looming threat response post-extinction training, a training paradigm was designed to update the looming stimulus with a fear association once again. Following completion of the extinction training paradigm, mice received an additional training day where they were exposed to the looming stimulus paired with a co-terminating 2 s electric foot shock (Fig 1f). When mice were tested 24 h later they showed a significant increase in freezing but not sheltering (Fig 1g–1i). Meanwhile, mice which were presented with either the looming stimulus or shock stimulus alone showed no significant increase in either freezing or sheltering (Figs 1i and S1h–S1m). As such, these results show that specific fear to the visual looming stimulus can be recovered by pairing the visual stimulus with an electric foot shock. Finally, to understand whether the rapid learning of fear following extinction was unique to the initially fearful looming stimulus, the paradigm was repeated with the nonthreatening dimming stimulus (Fig 1j). Salience of the dimming stimulus was confirmed by quantification of rearing, with a significant increase in rearing following dimming stimulus presentation on test 1 but not on test 2 or test 3 (S1k–S1l Fig). Mice did not display fear responses to the dimming stimulus on either test 1, test 2, or test 3, indicating that stimuli which are not initially fearful do not become associated with fear later in the extinction-updating paradigm (Fig 1j–1m). As such, we may conclude that the initial innate fear of the looming stimulus contributes to the learned fear association later.

## SCd and PAGdm responses attenuate with extinction and do not recover following updating with shock association

Defensive behavioral responses to visual looming stimuli are known to be mediated by the accumulation of threat information in the deep layer of the SC (SCd), which triggers the dorsal PAG (PAGdm) to initiate a behavioral response [4]. To understand how activity in these regions changes over the course of extinction and updating, in vivo fiber photometry recordings were performed. The calcium indicator GCaMP6f was expressed via local injection of AAV9-Syn-GCaMP6f-WPRE-SV40, and calcium-dependent fluorescence was recorded via an optical fiber implant at the injection site through which 465 and 405 nm excitation light were simultaneously delivered. Recordings were performed in either the SCd or the PAGdm (Figs 2 and S2, S3). During innate looming responses in Test 1, visual looming stimuli reliably drive increases in activity of the SCd. However in Test 2, following extinction, responses of the SCd are diminished and these responses do not recover during Test 3 following updating (Fig 2c–2e), although there may be a partial recovery in the first few seconds. A fast Fourier transform of the signal shows a strong peak at 1 Hz, coinciding with the frequency of looming stimulus presentation on Test 1, but not on Test 2 or 3 (S3d Fig). As such, extinction training results in an attenuation of both looming threat responses and activity of the SCd. Meanwhile, updating training does not recover SCd responses to visual looming stimuli, despite recovery of the behavioral responses. When mice undergo the "shock only" paradigm, wherein the shock

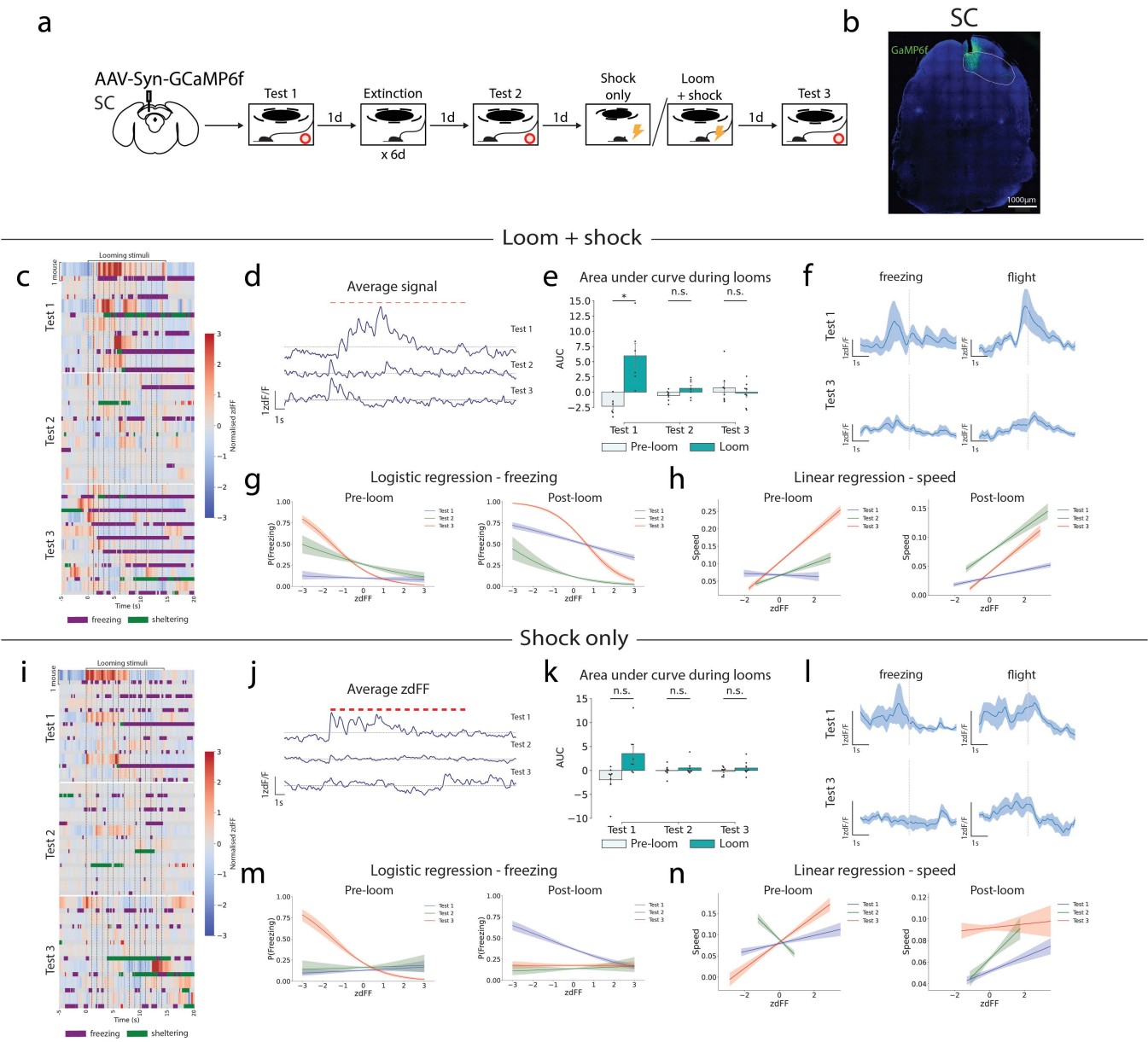

**Fig 2. Extinction attenuates SC and PAG looming stimulus responses which do not recover with updating training. (a)** GCaMP was virally expressed in the deep SC. Fiber photometric recordings of GCaMP fluorescence were taken over the course of the extinction-updating training paradigm. **(b)** Example histology from slices expressing GCaMP in the deep SC. **(c)** Heatmap with each blue-red row indicating zdFF values from individual mice during looming stimulus presentation. Beneath each zdFF row, a bar indicates whether the individual mouse was detected as freezing (purple) or sheltering (green). Looming stimulus onset is indicated by the dashed black lines, with first stimulus at time = 0 s. Each row is the response from a single mouse. **(d)** Mean traces of calcium activity in the deep SC on each test day (*n* = 6 mice). **(e)** Area under curve of GCaMP signal 15 s pre- and post-stimulus onset, for each day (Student *t* test). **(f)** Mean zdFF signal (shaded area = standard error of the mean) across bouts of freezing and flight behavior in all mice on test 1 and 3. Each behavior is initiated at time = 0 s. **(g)** Logistic regression of freezing probability with GCaMP signal, 15 s pre- and post-stimulus onset, for each test day. The shaded area is the 95% confidence interval. **(h)** Linear regression of speed with GCaMP signal, 15 s pre- and post-stimulus onset, for each day. The shaded area is the 95% confidence interval. **(i–n)** As in c–i, but for mice undergoing the "shock only" updating paradigm (*n* = 8 mice). n.s. *p* ≥ 0.05, **\**p* < 0.001. Details of all statistical comparisons may be found in S1 Data. Underlying raw data may be found in S2 Data.

is presented without the looming stimulus on the updating training day, there is no recovery of either behavioral responses or SCd activity (Fig 1i–1n).

The trends of SCd responses are mirrored in the PAGdm. The PAGdm shows a strong response to visual looming stimuli on test 1, which attenuates by test 2, and remains suppressed on test 3 (S2a–S2e Fig). Again, a fast Fourier transform of the signal on test 1 has a strong peak at 1 Hz for Test 1, indicating that the activity is in synchrony with looming stimulus presentation, but not on Test 2 or 3 (S3f Fig). As such, responses of the PAGdm are also diminished over the course of extinction training, and do not recover with updating despite recovery of the behavioral responses. The separation of SCd and PAGdm activity and looming behavioral responses, particularly freezing, are reflected by changes in correlation between these measures. On Test 1, a logistic regression of SCd activity with freezing reveals a linear, negative correlation. This negative linear correlation persists until Test 2, but on Test 3 an inverse sigmoidal relationship develops, indicating that SCd activity is suppressed during freezing (Fig 2g). Meanwhile, the SCd maintains a positive linear correlation with speed during looming across all test days (Fig 2h). This finding contrasts with mice that underwent "shock only" treatment during updating training. Their activity on Test 3 shows no correlation with freezing behavior (Fig 2m), suggesting that the inverse relationship between freezing and SC activity is unique to the mice which have the shock stimulus paired with looming stimuli.

Dorsal PAG activity on test 1 is positively correlated with freezing during stimulus presentation, but not on test 2 and test 3 (S2h Fig). As in the SCd, the PAGdm develops an inverse sigmoidal relationship with freezing on Test 3. Meanwhile, PAGdm activity is positively correlated with speed, albeit not to the same degree as in the SC (S2i Fig). These findings indicate that SCd and PAGdm activity is attenuated with extinction learning. This activity is not recovered following updating with fear, highlighting a lasting effect of extinction learning in these regions and suggesting that other brain regions may be responsible for the recovered behavioral response.

In summary, this data reveals a separation of threat responses to visual looming stimuli from SCd and PAGdm activity as a result of extinction-updating training. Therefore, another brain region must be implicated in the recovered threat responses of mice to the visual looming stimulus.

## c-Fos expression patterns reveal persistent activity of a vCA1 ensemble following extinction-updating

To investigate the activity of a wider range of brain areas in response to looming stimuli, c-Fos expression was analyzed throughout the brain. TRAP2xAi32 mice [35,36] were used to tag neurons in a c-Fos-dependent manner throughout the whole brain (Fig 3a). TRAP2xAi32 mice underwent one of three training paradigms; re-exposure versus no re-exposure, to determine how naïve looming alters c-Fos expression; re-exposure following extinction training versus a control training, to determine the impact of extinction training; or re-exposure following extinction-updating versus a control training to determine how fear updating further changes c-Fos expression (Fig 3b). Tagging was performed by delivering 4-OHT during naïve loom presentation, capturing the activity during innate loom responses. Following completion of the respective paradigm, mice were perfused 60 min after looming stimulus presentation. Neurons positive for EYFP are those active during innate loom responses, neurons positive for endogenous c-Fos are those active during loom responses at the completion of the paradigm, and neurons which are positive for both EYFP and endogenous c-Fos are responsive during both loom presentations (Fig 3b, 3c).

Following re-exposure to looming stimuli, no significant difference in overall c-Fos expression throughout the SC, PAG, or hippocampus was observed when compared with no re-exposure mice (Fig 3d). There was, however, a significant increase in the reactivation of previously labeled innately responsive cells in both the SCs and the vCA1 (Fig 3e). This indicates that sparse ensembles in both regions contribute to the innate coding of the visual looming response. Following extinction training, there was a significant decrease in overall c-Fos expression in the deep SC (SCd), dorsolateral PAG (PAGdl), and lateral PAG (PAGl) compared with nonextinction controls (Fig 3f). Meanwhile, the reactivation of innately responsive loom cells remains consistent throughout the SC, with a significant decrease in the PAGdm (Fig 3g). As such,

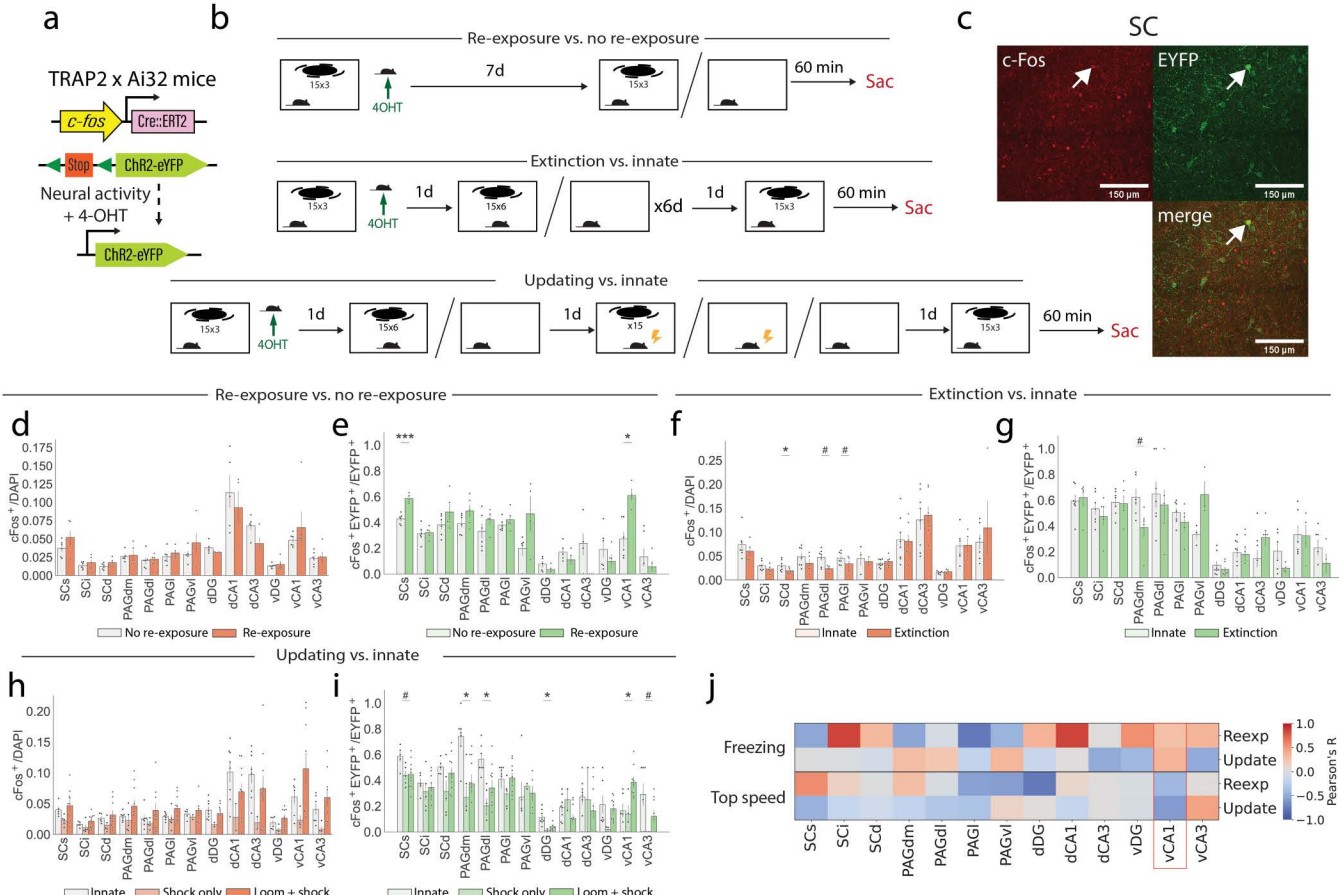

**Fig 3. Patterns of c-Fos expression in the SC, PAG, and HPC reveal a loss of activity in the SC and PAG following extinction-updating but persistent activity of the vCA1. (a)** Engram labeling strategy. **(b)** TRAP2xAi32 mice underwent one of three types of behavioral paradigms; re-exposure ($n = 6$ mice) vs. no re-exposure ($n = 6$ mice), extinction ($n = 7$ mice) vs. innate ($n = 8$ mice), or updating ($n = 11$ mice) vs. innate ($n = 8$ mice). Mice were injected with 4-OHT to transiently induce c-Fos dependent EYFP upon first exposure to the looming stimulus, and later perfused 60 min following stimulus presentation at the end of the behavioral paradigm. **(c)** Example histology from the SC, with an EYFP+c-Fos+ cell highlighted by the white arrow. **(d)** c-Fos expression upon completion of the re-exposure vs. no re-exposure paradigm (Student unpaired $t$ test with False discovery rate correction). **(e)** Proportion of initially tagged EYFP+ cells which also express c-Fos upon completion of the re-exposure vs. no re-exposure paradigm (Student unpaired $t$ test with False discovery rate correction). **(f and g)** As in d-e, but for the extinction vs. innate paradigm. **(h and i)** As in d and e, but for the updating "loom + shock" vs. "shock only" vs. innate paradigm (ANOVA with False discovery rate correction). **(j)** Pearson's correlations between each brain region and time spent freezing or top speed, during either the re-exposure, loom only, or loom+shock updating paradigm. n.s. $p \geq 0.05$, #$p < 0.05$, *$q < 0.05$, **$q < 0.01$, ***$q < 0.001$; where $p$ is the $p$ value, and $q$ is the false discovery rate-adjusted $p$ value. Details of all statistical comparisons may be found in S1 Data. Underlying raw data may be found in S2 Data.

while there is a decrease in the overall activity of several midbrain regions, the activity of specific loom-responsive ensembles in the SC and hippocampus remains equal to innate controls. Following updating training, overall c-Fos levels in the SC and PAG of mice undergoing the loom+shock paradigm, but not the shock only paradigm are consistent with innate controls, suggesting a recovery of c-Fos expression levels (Fig 3h). This aligns with the partial recovery of calcium activity detected in the loom + shock cohort only, perhaps indicating an increase of plasticity or partial recovery of these regions. There is a significant decrease in the reactivation of innately loom-responsive cells in the superficial SC (SCs), dorsomedial PAG (PAGdm), PAGdl, and dorsal DG (dDG). Meanwhile, there is a significant increase in the reactivation of vCA1 cells which were active during innate loom responses (Fig 3i). This suggests that updating training specifically results in

an attenuation of specific loom-responsive ensembles in the midbrain, but an upregulation of loom-responsive ensembles in the vCA1. Finally, when the reactivation of each brain region is correlated with freezing or top speed, the vCA1 is the only region which demonstrates a similar correlation between behavior and c-Fos reactivation in both the re-exposure and updating paradigms (Fig 3j).

Together, these data demonstrate that the innate visual looming response increases activity of specific ensembles in the SC and ventral CA1. Moreover, following extinction there is reduced activity in the SC and PAG, while the ventral CA1 is equal to no-extinction controls. Following fear updating, there is a recovery of overall c-Fos expression in the SC and PAG, but the activity of initially tagged innate loom-responsive cells does not recover. Meanwhile, there is increased reactivation of the initially tagged ventral CA1 loom-responsive cells. This suggests that the initially active vCA1 ensemble may also be implicated in the later updated threat responses to the visual looming stimuli in the absence of midbrain-driven responses.

**Loom-responsive cells of the vCA1 are necessary for learned but not innate defensive behavioral responses**

Based on the increased reactivation of innate-loom tagged vCA1 ensembles during both innate and updated loom threat response, alongside the established role of the vCA1 in fear conditioning reinstatement [26], the role of the vCA1 engrams in visual looming fear updating was investigated. To determine whether specific loom-responsive cells in the vCA1 are necessary in the updating visual looming fear response, the f-FLiCRE labeling system [37,38] was used. A viral cocktail containing AAV-Nrxn3b-Nav1.6-MKII-f-hLOV1-TEVcs(ENLYFQ/M)-tTA-VP16, AAV-GFP-CaM-uTEVp, and AAV-TRE:mCherry-p2a-eNpHR was locally injected into the vCA1. Optical fiber implants were used to deliver light to the vCA1. This labeling paradigm relies on intracellular calcium alongside blue light (473 nm) to induce labeling of neurons using the tetracylcine transactivator (tTA), inducing expression of the destination construct (Fig 4a–4c ). As a result, this system produces activity-dependent labeling during a dynamic window with a temporal precision on the order of seconds, and as previously demonstrated can be used to accurately label ensembles associated with very brief bouts of behavior [38].

To investigate whether specific ensembles activate during updating training are necessary for updated fear responses, vCA1 ensembles were tagged either pre-stimulus ("context label") or during loom and shock presentation ("loom label") of the updating training session (Fig 4d), inducing the expression of inhibitory Halorhodopsin. Twenty-four hours later, mice were exposed to two rounds of looming stimulus presentation; first with laser stimulation at 595 nm to inhibit tagged cells and second with no light-induced inhibition. For cells tagged during "context labeling", there was a significant increase in freezing following loom-presentation regardless of whether the laser was on (test 3A) or off (test 3B) (Fig 4e and 4f). However, for mice with cells tagged during loom and shock presentation, there was no increase in freezing whilst these cells were inhibited, but a significant increase in freezing when mice were presented the stimulus again without inhibition 90 s later (Fig 4g and 4h). There was no significant difference in the number of cells tagged during these two sessions (S5a–S5c Fig), despite their differing effect on behavior. This indicates that there is a specific ensemble of cells in the ventral CA1 which is activated specifically during presentation of the looming and shock stimuli, which is necessary for recall of the updated fear 24 h later.

Since looming and shock stimulus are presented together during updating, this ventral CA1 ensemble may contain both loom-responsive and shock-responsive cells. However, histological analysis of c-Fos expression revealed an increase in loom-responsive cells tagged during naïve presentation (Fig 3d–3i). Because of this, we next investigated the functional significance of these loom-responsive ensembles of the vCA1 in the updated behavioral response. Loom-responsive cells in the ventral CA1, or context-labeled controls, were tagged using the f-FLiCRE system in naïve mice. Then, mice underwent the extinction-updating training paradigm as before, except in this case each test day contained two rounds of looming stimulus presentation. During the first round, loom-responsive cells were optogenetically inhibited, and during the second round there was no inhibition applied (Fig 4i). The inhibition of context-labeled cells had no impact on looming fear responses on test 1, test 2, or test 3 (Fig 4j, 4k). For loom-labeled cells, there was no impact of inhibition on test 1 or

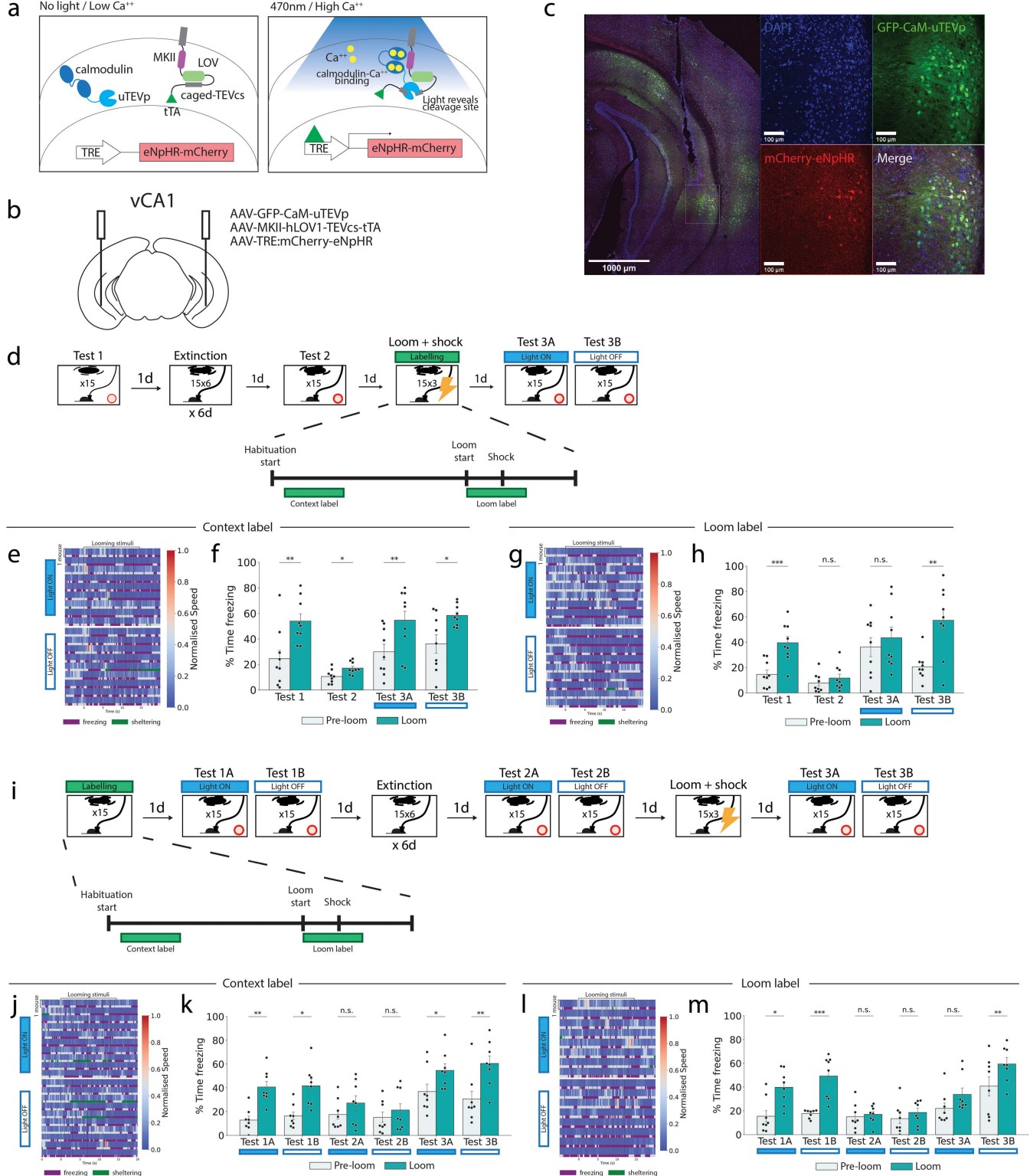

**Fig 4. Loom-responsive cells in the ventral CA1 are necessary for updated but not innate visual looming fear responses. (a)** f-FLiCRE labeling system. The coincident presence of calcium and blue light enables expression of eNPHR-mCherry. **(b)** Viral constructs for the f-FLiCRE labeling paradigm were injected into the ventral CA1. **(c)** Example histology of vCA1 f-FLiCRE labeling. Cells infected with the f-FLiCRE system will express GFP,

while cells which are tagged during labeling will express mCherry. **(d)** Behavioral timeline to investigate the role of ventral CA1 ensembles that are active during context presentation alone (context labeling) and during looming and shock stimuli presentation (loom labeling). **(e)** Heatmap indicating speed and behavioral responses during the light-on and light-off phases of test 3 for the context-labeled mice. **(f)** Freezing responses pre- and post-looming stimulus on each test day for the context labeled cohort (repeated measures ANOVA with post-hoc comparisons; $n = 9$ mice). **(g)** Heatmap indicating speed and behavioral responses during the light-on and light-off phases of test 3 for the loom-labeled mice. **(h)** Freezing responses pre- and post-looming stimulus onset on each test day for the loom-labeled cohort (repeated measures ANOVA with post-hoc comparisons; $n = 9$ mice). **(i)** Behavioral timeline to investigate the role of ventral CA1 ensembles that are active during either during context presentation (context labeling) or during naïve looming stimulus presentation (loom labeling). **(j)** Heatmap indicating speed and behavioral responses during the light-on and light-off phases of test 3 for the context-labeled mice. **(k)** Freezing responses pre- and post-looming stimulus on each test day for the context labeled cohort (repeated measures ANOVA with post-hoc comparisons; $n = 8$ mice). **(l)** Heatmap indicating speed and behavioral responses during the light-on and light-off phases of test 3 for the innate loom-labeled mice. **(m)** Freezing responses pre- and post-looming stimulus on each test day for the innate loom-labeled cohort (repeated measures ANOVA with post-hoc comparisons; $n = 8$ mice). n.s. $p \geq 0.05$, *$p < 0.05$, **$p < 0.01$, *** $p < 0.001$. Details of all statistical comparisons may be found in S1 Data. Underlying raw data may be found in S2 Data.

2. However, on Test 3 following updating, there was no change in freezing levels while loom-responsive cells were inhibited, but there was a significant increase in freezing to the looming stimulus 90 s later when these cells were no longer being inhibited (Fig 4l and 4m). As such, there are cells in the ventral CA1 which are active during innate visual looming responses but are not required for innate responses. However, these same cells are later necessary for recall of the updated visual looming fear responses.

## Discussion

In this study, we establish a novel behavioral paradigm to explore how the brain facilitates adaptive responses to a highly innate, threat-related visual stimulus. We show that the innate visual looming response is subject to a long-lasting and contextually generalizable extinction. However, threat responses to the visual stimulus can be recovered via pairing of the stimulus with an electric footshock. During extinction training, activity in the deep superior colliculus (SCd) and dorsal periaqueductal gray (PAGdm) attenuate and do not recover after updating. Furthermore, by analyzing c-Fos expression in the midbrain and hippocampus, we have identified a population of vCA1 cells which respond to looming stimuli during naïve presentation, in addition to specific loom-responsive ensembles in the SC. Over the course of extinction-updating, the specific loom-responsive cells of the SC no longer respond to the visual looming stimuli and this does not recover with updating. Only the initially tagged vCA1 ensembles show a recovery of their response to looming stimuli following updating training. Functional manipulation of these vCA1 ensembles revealed that these cells specifically respond to looming stimuli and are necessary for adaptive behavioral responses following updating. This study revealed that the extinction and updating of innate fear responses to a looming stimulus rely on hippocampus-dependent mechanisms, expanding our understanding of hippocampal function into ethologically relevant behaviors. These findings demonstrate a novel adaptive pathway for the generation of innate behavioral responses while still allowing for behavioral flexibility and learning.

### Adaptability of the innate looming response

Our visual looming extinction-updating paradigm expands upon literature indicating that the innate looming response is adaptable. Prior studies show that spatial memory modulates flight behavior via retrosplenial cortex-SC projections [39]. Mice can also remember areas where looming stimuli were previously presented to later avoid those areas [4], as well as learning to fear a tone previously paired with looming stimuli (albeit to a lesser degree than with an electric shock) [40]. Additionally, mice can learn to reduce or abolish responses to the looming stimulus, highlighting stimulus-specific adaptation [14–17]. Our paradigm uniquely demonstrates a transition from looming-related fear to complete behavioral attenuation followed by the updating of fear responses to the same looming stimulus, permitting within-subject comparisons of innate versus learned fear responses. Crucially, presentation of the reinforcing stimulus on its own is not sufficient to recover fear responses. Additionally, an alternative visual stimulus which is nonfearful cannot become associated with fear

under the same repeated-exposure paradigm, indicating that a memory of the initial fear towards the looming stimulus must contribute to the rapidly learned fear association following updating.

The extinction phenotype characterized here has some distinctions from classical fear conditioning extinction. For example, it persists at least 3 weeks after extinction training and generalizes to other contexts. As such, one may consider this a very persistent extinction memory. This phenomenon is also distinct from sensory habituation in several ways. Firstly, c-Fos analysis revealed continued responsiveness of specific loom-responsive ensembles in the superior colliculus post training, while habituation typically evoked a reduced activity in sensory processing regions [41,42]. Secondly, the ability of the mouse to learn an association between the visual looming stimulus and an electric foot shock demonstrates continued detection of the visual looming stimulus, even in the absence of a behavioral response. And thirdly, while a habituated response can typically be recovered by presentation of a startling stimulus on its own, an electric foot shock on its own is not sufficient to recover behavioral responses to the visual looming stimulus. Meanwhile, an electric foot-shock paired with the visual looming stimulus was sufficient to recover fear responses to visual looming stimuli. This learning occurred with only a single pairing of the looming stimulus with the shock, so is learned quite readily despite the otherwise persistent extinction memory.

## Distinctions from classical conditioning

In classical fear conditioning, an initially neutral and typically novel stimulus becomes associated with fear by pairing it with an unconditioned stimulus, such as an electric footshock [30]. The reinstatement paradigm presented here is distinct, due to the previously existing innate fear association and subsequent extinction of the looming stimulus. Indeed, when an alternative visual "dimming" stimulus that is not initially fearful is used in the same training conditions as our reinstatement paradigm, no association between the novel stimulus and the footshock is observed (Fig 1m). Thus, the innate threat history of the visual stimulus is vital for the footshock to elicit fear behavior.

This interpretation is supported by the demonstrated functional role of vCA1 ensembles that were responsive during innate looming fear responses. When these initially innate loom responsive cells are inhibited following association of the looming stimulus with a footshock, the mice do not demonstrate fear behavior. As such, it is a vCA1 ensemble-dependent memory of the innate fear of the looming stimulus that is allowing for the rapid re-association with fear following extinction. In contrast, the dimming stimulus is not innately fearful and so fear is not as readily learned following so many exposures.

## A novel role for the hippocampus in adaptive looming fear responses

We identify a novel role for vCA1 in looming-evoked behavior. Alongside canonical SC and PAG circuits, sparse vCA1 cells form looming-responsive engrams that are necessary for learned but not innate defensive behavioral responses to looming stimuli. These vCA1 loom-responsive engram cells are not necessary for evoking innate defensive behavioral responses, but do become necessary for learned responses following the extinction-updating paradigm. Meanwhile, the SC and PAG responses which are known to drive visual looming fear responses at a population level [4] become much less responsive to visual looming stimuli and in fact have an inverse relationship with freezing. As such, it seems that the behavioral response to visual looming stimuli has transitioned from SC-dependency to vCA1-dependency. The vCA1 is implicated in a wide range of emotionally salient, rapidly forming memories, including fear conditioning [25–27,43], social memory [44], and reward [43,45]. Therefore, these loom-responsive stimuli in the vCA1 may aid in adaptation of behavior in response to looming stimuli, such as avoiding areas where looming stimuli were previously presented [4,13].

Additionally, this study expands our understanding of hippocampal function beyond learned and contextual behaviors and into the domain of innately salient and ethologically relevant behaviors. In addition to innately salient same and opposite sex social stimuli [44,46] and thermogenesis-inducing cold exposure [47], this study demonstrates that innately threatening stimuli are represented in the hippocampus in a manner which directly contributes to future responses to the stimuli. This role is supported by the observation of that hippocampal lesions result in reduced defensive behaviors

towards threatening stimuli in rats and primates [48–51], and the extensive. Given this, it may be possible that the hippocampus directly contributes to adaptability and flexibility of behavioral responses to innately salient stimuli [52], warranting further investigation into the contribution of specific hippocampal ensembles in innate behavior.

### Insights into instinct and memory interactions

In summary, the SC and PAG mediate rapid, defensive looming responses, reflecting the evolutionary importance of these circuits. However, the brain balances this hard-wired response with adaptive flexibility, storing looming-related information in the hippocampus to accommodate new experiences. This strategy likely reflects selection pressures on rapid, innate threat responses where immediate action is critical and learning opportunities are minimal. By recruiting vCA1 for adaptive learning, the brain can accommodate plasticity of behavior without compromising rigid survival circuits. Additionally, when updating of innate midbrain circuits is essential, as is the case in the extinction learning paradigm, the vCA1 allows for long term storage of the fear of the looming stimulus which is later essential for the rapid recovery of fear. Thus, the looming response exemplifies how even highly innate, survival-critical behaviors maintain plasticity, blending instinct with the flexibility afforded by memory systems.

## Materials and methods

### Experimental model details

**Animals.** All wild-type behavior, fiber photometry, and f-FLiCRE experiments were performed with C57BL/6J mice. Whole-brain engram labeling experiments were performed with TRAP2 x Ai32 [35,36,53]. Mice were housed in groups of 3–6 in cages with enrichment, food, and water ad libitum under a 12 h light/dark cycle. The animal room was kept at a constant temperature of 22°C and 50% humidity. Mice used for experiments were approximately 8–12 weeks of age. All experiments were performed during the light phase. All experimental manipulations were carried out accordingly with Health Products Regulatory Authority (HPRA) Ireland guidelines and the principle of the 3R's (https://nc3rs.org.uk/) and with ethical approval from the Trinity College Dublin Animal Research Ethics Committee and from the Health Products Regulatory Authority (HPRA; Project Authorisation Number AE19136/P165).

### Method details

**Visual looming presentation.** The looming behavioral arena consisted of a rectangular floor (46 cm width × 30 cm length) and a computer monitor on top (28 cm above). A red polycarbonate tube (7.5 cm diameter × 15 cm length) was positioned at the center of the rear wall to provide a shelter. One day prior to experiments, mice were habituated to the arena for 5 min while the monitor displayed a white image. On experimental day, after 5 min of acclimation with a white monitor, the looming stimulus was triggered when the animals were located in the half of the arena opposite the shelter. The stimulus consisted in black disc appearing in the center of the monitor at a diameter of 2° of visual angle, expanding to 20° in 250 ms, remaining at that size for 250 ms before disappearing. This was repeated 15 times with 500 ms between repetitions. After 15 presentations, the mice were left in the arena with a white monitor for 1 min before being taken back to their home cage. The arena and the shelter were cleaned with Trigene between animals. Custom Python code using the PsychoPy library [54] was used to present stimuli and record video of the animals (available at: https://doi.org/10.5281/zenodo.17048095). Time spent freezing and sheltering was scored using DeepLabCut tracking [55], tracking a total of 14 body parts. Sheltering was defined as having all body parts (excluding the tail) within the y-axis limits of the shelter, and at least half of all body parts (excluding the tail) contained within the x-axis limits of the shelter. Freezing was defined as the cessation of all movement except breathing [56,57]. Quantitatively, this was defined as periods in which no more than 1 body part was detected as moving a distance of 2 mm or greater, for a time period of at least 0.5 s. DeepLabCut behavioral scoring was verified by manual scoring in all wild-type experiments and this was used to define the aforementioned parameters.

**Visual looming extinction.** After habituation day and looming test 1 described above, mice were repeatedly exposed to three waves of 15 looming stimulus presentations, separated by 1 min each, or a white screen control for the same length of time on 6 consecutive days in the absence of the shelter. On the following days, mice underwent looming test 2, with the same parameters as test 1, the shelter being present.

**Visual looming updating.** This paradigm was performed in a MedAssociates fear conditioning chamber to allow the application of a footshock. Visual stimuli were presented via a monitor as described above, beginning at a diameter of 2° of visual angle, expanding to 20° in 250 ms, remaining at that size for 250 ms before disappearing. This was repeated 15 times with 500 ms between repetitions. After completing the extinction paradigm in the MedAssociates chamber in the same manner as above, mice received an additional training day without a shelter. After a 5 min habituation period, mice were either presented with a train of 15 looming stimuli paired with a co-terminating 0.75-mA 2 s foot-shock, or just the foot-shock without visual stimulus presentation. Mice were then tested 24 h later with a shelter.

**Visual looming novel context post-extinction.** After completing the extinction paradigm, mice were tested in a novel context. This involved the use of an arena in a different room with distinct spatial dimensions (although the same proportion of the visual field was used for looming stimulus generation), distinct tactile cues, patterns on the walls, and distinct olfactory cues.

**Visual dimming stimulus presentation.** Custom Python code using the PsychoPy library [54] was used to present stimuli and record video of the animals (available at: https://doi.org/10.5281/zenodo.17048095). The visual dimming stimulus consisted of a disk of constant radius of 20° of the visual angle and gradually increasing darkness. The disk increased from a contrast of 0% to 75% (psychopy disk color = −0.5) over a period of 250 ms on a white background, and maintained a contrast of 75% for 250 ms before disappearing. This was repeated 15 times with 500 ms between repetitions. Parameters were chosen based on preliminary experiments to be nonthreatening (absence of freezing and flight responses) but salient. Saliency was determined by the presence of rearing behavior. Rearing was scored manually and defined as period in which both front paws were raised off the ground without being used to lean on the wall or shelter or being engaged in grooming behavior.

**Stereotactic surgery.** Stereotactic surgery was performed at least two weeks before starting any behavioral procedures. Mice were anaesthetized with 500 mg kg$^{-1}$ avertin and head fixed on the stereotaxic instrument. Ophthalmic ointment was applied to eyes in order to prevent drying and intracutaneous lidocaine was injected in the skull. The surgical area was then washed with chlorohexidine and 70% ethanol three times before using a blade to perform a 1–1.5 cm incision through the midline of the scalp. After swabbing with increasing concentrations of ethanol, hydrogen peroxide was used to enhance visualization of lambda and bregma. Animals underwent bilateral craniotomy and holes were drilled at the appropriate coordinates for the target region, detailed below. The viral mix was injected using a micro syringe pump (UMP3; WPI) and a 10 µL Hamilton syringe (701LT; Hamilton) at the appropriate depth for the target region. Injection of virus was at a rate of 60 nL/min. After completion of the injection, the needle was left in the injection site for an additional 5 min. For optogenetic experiments, a fiber implant (Doric) was fixed to the stereotaxic adapter and was lowered in the brain, 0.1–0.2 mm above virus injection site. To secure the implant, an even layer of Metabond was applied and let dry for about 10–15 min. Dental cement was then used to secure the cap to the implant. The protective cap was provided by cutting the top portion of a black polypropylene microcentrifuge tube. Animals were then left in a warm box (29°C) until recovered from anesthesia and then were returned to their homecage. On the day of the surgery and on the two following days mice were injected subcutaneously with meloxicam (0.075 mL/5 g). Mice were allowed to recover for 14 days before any behavioral procedure was performed. Recovery was monitored with the use of a HPRA-approved recovery assessment table.

**Viruses.** pAAV.Syn.GCaMP6f.WPRE.SV40 was a gift from Douglas Kim & GENIE Project (Addgene plasmid # 100837-AAV9; http://n2t.net/addgene:100837; RRID:Addgene_100837)

Plasmids for AAV-TRE:mCherry-p2a-eNpHR, AAV-GFP-CaM-uTEVp, and AAV-Nrxn3b-Nav1.6-MKII-f-hLOV1-TEVcs(ENLYFQ/M)-tTA-VP16 were a gift from Alice Ting. Viral vectors were produced and purified by Mark Brimble and Stephen Winston as previously described [38]. The viral vectors used in this study are summarized in Table 1.

**In vivo fiber photometry.** A single-site, two-color fiber photometry system (Doric Lenses) was used for in vivo fiber photometry, adapting the protocol of Silva and colleagues [59]. Recordings were taken of the 405 nm isosbestic calcium-independent signal and the 465 nm calcium-dependent GCaMP6f fluorescence. The LED power was constant across animals and experimental sessions (200 mW). The presence of visual looming stimuli was synchronized with fluorescence recordings via a transistor-transistor logic pulse from an Arduino module. Animals were habituated to patch chords in a familiar environment (not recorded) and in the looming environment (recorded) for 5 min each over the 2 days prior to the commencement of the extinction-updating training paradigm.

Acquired data files were processed using code adapted from Martianova and colleagues [60]. A moving mean was applied to smooth the signal. And baseline correction was performed using the adaptive iteratively reweighted Penalized Least Squares (airPLS) algorithm to remove the slope and low frequency fluctuations in the signal. Each signal was normalized according to the mean and standard deviation. Using nonnegative robust linear regression, the normalized 405 signal was fit to the 465 signal, in the format $y = a \times x + b$. The parameters of this regression ($a,b$) were used to calculate new values for the 405 signal fitted to the 465 signal. Finally, the normalized fluorescence $dF/F$ ($z\ dF/F$) is calculated as the normalized 465 signal subtracting the fitted normalized 405 signal.

**TRAP2xAi32 engram labeling.** TRAP2xAi32 were placed in the visual looming assay for a 5 min acclimation period before presentation of any stimuli. Three rounds of 15 looming stimuli were presented at a frequency of 1 loom per second, with a 1 min rest period between rounds. Immediately following the end of the exposure paradigm, mice were intraperitoneally injected with 4-hydroxtamoxifen (4-OHT; Sigma Aldrich) at a dosage of 50 mg/kg to transiently induce labeling in cells in a c-Fos-dependent manner.. All 4-OHT was freshly prepared on the labeling day in a mix of sunflower seed oil and castor oil (4:1, Sigma-Aldrich) at 10 mg/ml.

**f-FLiCRE engram labeling and stimulation.** Mice were injected with a 1:1:1 viral cocktail including AAV-TRE:mCherry-p2a-eNpHR, AAV-GFP-CaM-uTEVp, and AAV-Nrxn3b-Nav1.6-MKII-f-hLOV1-TEVcs(ENLYFQ/M)-tTA-VP16 in the ventral CA1. This viral cocktail allows expression of the f-FLiCRE labeling paradigm, which will result in expression of the inhibitory halodrhodopsin eNpHR upon coincidence of blue light (473 nm) and calcium. For tagging of cells during the updating training session, mice initiated the extinction-updating training paradigm 3 weeks post-surgery. On the updating training session, the blur laser was turned on for a 75 s period either beginning 1 min into the acclimation session, or at the onset of visual stimulus presentation. Light stimulation was delivered via continuous wave at a power of 5 mW. The following day, mice underwent 2 rounds of visual stimulus presentation separated by at least 90 s. On the first round, a yellow laser (595 nm) was turned on to inhibit cells from 30 s pre-stimulus until 30 s post-stimulus onset. Laser stimulation was performed at 10 Hz with 20 ms pulses, at a power of 5 mW. On the second round, no light was turned on.

**Table 1. Summary of viral vectors.**

| Virus | Addgene ref | Serotype | Titer | Dilution | Paper reference |
|---|---|---|---|---|---|
| AAV9-Syn-GCamp6f | #100,837 | AAV9 | ≥1 × 10¹³ vg/mL | 1 in 2 | Chen and colleagues [58] |
| AAV-TRE:mCherry-p2a-eNpHR (f-FLiCRE 1) | #158,703 | AAV-DJ | ≥1 × 10¹³ vg/mL | Undiluted | Kim and colleagues [37] |
| AAV-GFP-CaM-uTEVp (f-FLiCRE 2) | #163,032 | AAV-DJ | ≥1 × 10¹³ vg/mL | Undiluted | Kim and colleagues [37] |
| AAV-Nrxn3b-Nav1.6-MKII-f-hLOV1-TEVcs(ENLYFQ/M)-tTA-VP16 (f-FLiCRE 3) | #163,031 | AAV-DJ | ≥1 × 10¹³ vg/mL | Undiluted | Kim and colleagues [37] |

For mice tagged during innate loom presentation, mice were presented with looming stimuli before any training commenced and tagged via the same protocol. Mice then commenced the extinction-updating training paradigm, with two rounds of visual looming stimuli, receiving yellow light inhibition for the first round and no inhibition for the second.

**Immunohistochemistry.** Mice were injected intraperitoneally with a terminal dose of Sodium Pentobarbital (50 µL) and perfused transcardially with PBS followed by 4% paraformaldehyde. Brains were then collected and left in 4% PFA at room temperature for 24 h, after which they were moved to PBS and stored at 4°C. Brains were cut using a vibratome into 50 µm coronal slices for all experiments except for whole brain imaging experiments, which were cut into 100 µm coronal slices.

For the immunostaining, slices were first washed in PBT 0.2% three times for 10 min. Then, they were incubated in a blocking solution (PBT 0.2% + NGS 10%) for 1 h at room temperature. They were incubated with primary antibody (in PBT 0.2% + NGS 3%) overnight at 4°C while shaking. On the following day, slices underwent three 10 min washes in PBT-0.2% before incubation with secondary antibody (PBT 0.2% + NGS 3%) for 2 h in the dark at room temperature. Then, after three additional 10 min washes in PBT 0.2%, slices were stained with DAPI (1:1000 in PBS) for 15 min at room temperature. Slices were finally mounted on superfrost slides using Vectashield-DAPI. The antibodies used for the c-Fos staining were rabbit anti-c-Fos (Synaptic Systems #226008 at 1:1,000) and anti-rabbit Alexa-568 (Invitrogen) (1:500). ChR2-EYFP was stained using a primary chicken anti-GFP IgY fraction (Invitrogen) (1:500), and rabbit anti-chicken Alexa-488 IgG (Invitrogen) (1:500).

**Image acquisition and analysis.** Images were taken on a laser confocal microscope (Leica SP8 gated STED) with a 20× magnification unless otherwise stated. At least 4 images were taken for each region of interest per mouse. When the implant was not targeted correctly into the region of interest the animal was excluded from subsequent analysis. For each region of interest, density of cells was estimated by sampling cell density in the region of interest to calculate the number of DAPI cells per $µm^2$. The total number of cells in a region was calculated by multiplying this value by the total area of the region counted. The number of c-Fos positive cells was manually counted using ImageJ.

## Quantification and statistical analysis

Statistical analysis was performed using the pingouin and scipy.stats packages on Python. Data was tested for normality using the Shapiro–Wilk test. An unpaired Student $t$ test was used to compare means of two independent groups that assume equal variance between groups, while the paired test was used for dependent groups. For data with more than two groups, an ANOVA was performed to determine if there was a significant difference. If so, and the data were normally distributed, post-hoc analyses were performed for pairwise comparisons. For data with multiple comparisons from the same subjects, False discovery rate correction via the Benjamini–Hochberg method was performed to calculate adjusted $q$ values.

## Supporting information

**S1 Fig. Looming extinction lasts at least 3 weeks, generalizes to other contexts, and does not recover with a shock or loom stimulus provided separately.** (**a**) Behavioral paradigm to compare looming responses with and without a shelter. (**b**) Freezing and sheltering behavior of mice 15 s pre- versus 15 s post-stimulus onset both with and without the presence of a shelter (Student $t$ test; No shelter, $n = 8$ mice; Shelter, $n = 9$ mice). (**c**) Heatmap indicating the speed of mice, with each blue-red row indicating the speed of a single mouse. Beneath the speed, a bar indicates whether the individual mouse was detected as freezing (purple) or sheltering (green). (**d**) Extinction-renewal assessment paradigm. (**e**) Freezing and sheltering behavior across test days in the extinction-renewal paradigm (Tukey's pairwise comparisons; ctrl, $n = 7$ mice; new context, $n = 7$ mice). (**f**) Extinction-spontaneous recovery assessment paradigm. (**g**) Freezing and sheltering behavior in the extinction-spontaneous recovery paradigm (Tukey's pairwise comparison; $n = 7$ mice). (**h**) Extinction- updating training paradigm with only the looming stimulus. (**i**) Freezing and sheltering responses of mice from the "loom only" cohort (Repeated

measures ANOVA and post hoc Student paired $t$ test; $n=9$ mice). (**j**) Heatmap indicating the speed of mice, with each blue-red row indicating the speed of a single mouse. (j) "Dimming" extinction-updating training paradigm. (**k**) Rearing responses of mice from the dimming "shock only" and dimming "dim + shock" cohorts in the dimming extinction-updating paradigm (Repeated measures ANOVA and post hoc Student paired $t$ test; $n=9$, 9 mice). n.s. $p≥0.05$, *$p<0.05$, **$p<0.01$, ***$p<0.001$. Details of all statistical comparisons may be found in S1 Data. Underlying raw data may be found in S2 Data.
(TIF)

**S2 Fig. Fiber photometry recordings of the dorsal PAG for mice undergoing the updating paradigm.** (**a**) GCaMP was virally expressed in the dorsal PAG. Fiber photometric recordings of GCaMP fluorescence were taken over the course of the extinction-updating training paradigm. (**b**) Example histology from slices expressing GCaMP in the dorsal PAG. (**c**) Heatmap with each blue-red row indicating zdFF values from individual mice during looming stimulus presentation. Beneath each zdFF row, a bar indicates whether the individual mouse was detected as freezing (purple) or sheltering (green). Looming stimulus onset is indicated by the dashed black lines, with first stimulus at time=0 s. Each row is the response from a single mouse. (**d**) Mean traces of calcium activity in the dorsal PAG on each test day ($n=7$ mice). (**e**) Area under curve of GCaMP signal 15 s pre- and post-stimulus onset, for each day (Student $t$ test). (**f**) Mean zdFF signal (shaded area = standard error of the mean) across bouts of freezing and flight behavior in all mice on tests 1 and 3. Each behavior is initiated at time=0 s. (**g**) Logistic regression of freezing probability with GCaMP signal, 15 s pre- and post-stimulus onset, for each test day. The shaded area is the 95% confidence interval. (**h**) Linear regression of speed with GCaMP signal, 15 s pre- and post-stimulus onset, for each day. The shaded area is the 95% confidence interval. n.s. $p≥0.05$, *$p<0.05$. Details of all statistical comparisons may be found in S1 Data. Underlying raw data may be found in S2 Data.
(TIF)

**S3 Fig. Mean responses of the SC and PAG across the extinction-updating training paradigm.** (**a**) Mean responses of the deep SC across all mice undergoing the loom + shock updating paradigm on each training and test day. (**b**) As in a, but for mice undergoing the shock-only paradigm. (**c**) As in a, but with recordings from the dorsal PAG. (**d**) Fast Fourier transform of the GCaMP signal in the deep SC of "loom ock" mice during stimulus presentation on each day. The most ethologically relevant peak occurs is at 1 Hz, which is the frequency of looming stimulus presentation. (**e–f**) As in (d), but for (e) the deep SC of "shock only" mice and (f) the dorsal PAG of "loom + shock" mice.
(TIF)

**S4 Fig. EYFP tagged cells in response to looming stimuli.** (**a**) Number of engram cells tagged during looming for the re-exposure ($n=6$ mice) vs. no re-exposure ($n=6$ mice) paradigm (Student paired $t$ test with False discovery rate correction). (**b** and **c**) As in a, but for (b) extinction ($n=7$ mice) vs. innate ($n=8$ mice), and (c) updating "loom + shock" ($n=11$ mice) vs. "shock only" ($n=5$ mice) vs. innate ($n=8$ mice) (ANOVA with False discovery rate correction). n.s. $p≥0.05$, #$p<0.05$, * $q<0.05$; where $p$ is the $p$ value, and $q$ is the false discovery rate-adjusted $p$ value. Details of all statistical comparisons may be found in S1 Data. Underlying raw data may be found in S2 Data.
(TIF)

**S5 Fig. TRAP2xAi32 behavior quantification.** (**a**) TRAP2xAi32 labeling method. (**b**) Freezing responses of TRAP2xAi32 mice undergoing the re-exposure or no re-exposure paradigm (Repeated measures ANOVA and post hoc Student paired $t$ test; $n=6$, 6 mice). (**c**) Freezing responses of TRAP2xAi32 mice undergoing the extinction or no-extinction paradigm (Repeated measures ANOVA and post hoc Student paired $t$ test; $n=12$, 13 mice). (**d**) Freezing responses of TRAP2xAi32 mice undergoing the updating "loom + shock", "shock only" or no-extinction no-shock re-exposure paradigm (Repeated measures ANOVA and post hoc Student paired $t$ test; $n=14$, 6, 6 mice). Details of all statistical comparisons may be found in S1 Data. Underlying raw data may be found in S2 Data.
(TIF)

**S6 Fig . f-FLiCRE labeling quantification.** (**a**) The proportion of DAPI cells which were infected with the f-FLiCRE construct ($n = 9, 9, 10, 8$ mice). (**b**) The proportion of infected cells tagged during stimulus presentation ($n = 9, 9, 10, 8$ mice). (**c**) The proportion of DAPI cells tagged during stimulus presentation ($n = 9, 9, 10, 8$ mice). Details of all statistical comparisons may be found in S1 Data. Underlying raw data may be found in S2 Data.
(TIF)

**S1 Data. Statistical summary.** A summary of all statistical tests performed and their outcomes.
(XLSX)

**S2 Data. Data summary.** Excel file containing the raw data points for the data used in this manuscript.
(XLSX)

## Acknowledgments

We thank Tamara Boto, Maximilian Jösch, Mani Ramaswami, and Ryan Lab members for useful scientific discussions and support. Inclusion and diversity: we support inclusive, diverse, and equitable conduct of research.

## Author contributions

**Conceptualization:** Paul B. Conway, Antoine Harel, James D. O'Leary, Mark A. Brimble, Gisella Vetere, Tomás J. Ryan.

**Data curation:** Paul B. Conway, Livia Autore, Andrea Muñoz Zamora, Zijun Wang, Arman A. Tavallaei, Gisella Vetere.

**Formal analysis:** Paul B. Conway, Livia Autore, Andrea Muñoz Zamora.

**Funding acquisition:** Tomás J. Ryan.

**Investigation:** Paul B. Conway, Livia Autore, Andrea Muñoz Zamora, Antoine Harel, Zijun Wang, Arman A. Tavallaei, Clara Ortega-de San Luis, James D. O'Leary.

**Methodology:** Paul B. Conway, Livia Autore, Andrea Muñoz Zamora, Antoine Harel, Arman A. Tavallaei, Stephen M. Winston, Clara Ortega-de San Luis, James D. O'Leary, Mark A. Brimble, Gisella Vetere, Tomás J. Ryan.

**Project administration:** Paul B. Conway, Tomás J. Ryan.

**Resources:** Paul B. Conway, Livia Autore, Andrea Muñoz Zamora, Stephen M. Winston, James D. O'Leary, Mark A. Brimble, Gisella Vetere, Tomás J. Ryan.

**Supervision:** Gisella Vetere, Tomás J. Ryan.

**Validation:** Gisella Vetere.

**Writing – original draft:** Paul B. Conway, Tomás J. Ryan.

**Writing – review & editing:** Paul B. Conway, Livia Autore, Clara Ortega-de San Luis, Gisella Vetere, Tomás J. Ryan.

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
