## [Editor Report · Decision Letter 0]

30 Jul 2025

Dear Tomás, 

Thank you for submitting your revised manuscript entitled "Plasticity of visual looming response reveals a dissociation of innate and learned components" for consideration as a Research Article by PLOS Biology.

Your manuscript has now been evaluated by the PLOS Biology editorial staff as well as by an Academic Editor with relevant expertise and I am writing to let you know that, based om the prior peer review history, your rebuttal letter and our Academic Editor's assessment of the revised manuscript, we are likely to publish your manuscript as a Short Report. 

We have a few additional suggestions and editorial requests that we would like you to implement already at this point, but there will likely be a few more requests after the full submission of the manuscript.

* We would like to suggest a different title to improve its accessibility for our broad audience: "Extinction and reinstatement of innate fear responses to a looming stimulus rely on hippocampus-dependent mechanisms"

* In your revision, please clearly highlight the conceptual advance of how your study expands our understanding of hippocampal function into the domain of ethologically relevant behaviors.

* We agree with the reviewers that the term reinstatement may be misleading and recommend adopting alternative terminology in the revised version.

* DATA POLICY:

Regardless of the method selected, please ensure that you provide the individual numerical values that underlie the summary data displayed in the following figure panels as they are essential for readers to assess your analysis and to reproduce it: 1DEHILM, 2K, 3DEFGHI, 4FKHM, S1BEGHL, S2E, S4ABC, S5BCD and S6ABC. 

* CODE POLICY

* Please note that per journal policy, the model system/species studied should be clearly stated in the abstract of your manuscript. 

* Please include the full name of the IACUC/ethics committee that reviewed and approved the animal care and use protocol/permit/project license. Please also include an approval number.

Only after you have submitted the revised manuscript including the meta data, we will be able to complete our editorial checks, so there may be a few more requests at the next stage. Your manuscript will also be seen again by the Academic Editor. 

To provide the metadata for your submission, please Login to Editorial Manager (https://www.editorialmanager.com/pbiology) within two working days, i.e. by Aug 13 2025 11:59PM.

Please upload the response to the reviews as a 'Prior Peer Review' file type, which should include the reports in full and a point-by-point reply detailing how you have addressed the reviewers' concerns. 

Kind regards,

Christian

Christian Schnell, PhD

Senior Editor

PLOS Biology

cschnell@plos.org

---

## [Editor Report · Decision Letter 1]

14 Aug 2025

Dear Tomás,

Thank you for your patience while we considered your revised manuscript "Extinction and subsequent updating of innate fear responses to a visual looming stimulus rely on hippocampus-dependent mechanisms" after review at another journal for publication as a Short Reports at PLOS Biology. This revised version of your manuscript has been evaluated by the PLOS Biology editors and the Academic Editor.

Based on our Academic Editor's assessment of your revision, we are likely to accept this manuscript for publication, provided you satisfactorily address the remaining points raised by the Academic Editor and address the following data and other policy-related requests:

* Please add the links to the funding agencies in the Financial Disclosure statement in the manuscript details.

* Please include the approval/license number of the ethical approval for the animal experiments.

* Please ensure that the figure legends in your manuscript include information on where the underlying data can be found.

* Please note that per journal policy, the model system/species studied should be clearly stated in the abstract of your manuscript. 

* Please include a dedicated section in the Discussion on the main issue raised by the reviewers during an earlier round of review at another journal, that is: the behavioral paradigm could be interpreted as a variation of classical fear conditioning, in which hippocampal involvement would be expected. We recognize that you have clarified that your paradigm allows a direct comparison between innate and learned responses to the same visual stimulus (looming disc), enabling you to trace how innate hippocampal representations are later reused during learned fear. Your data suggest that loom-responsive ensembles tagged during innate behavior are reactivated during learned freezing, and that these specific ensembles, rather than generic contextual ones, are necessary for the behavioral response.

You have included this somehow in your response to the reviewers, but we would like to see a specific section on this matter, ideally tagged as (suggestion) "Is extinction and subsequent updating of innate fear a form of classical conditioning?" We think that it is very important that the revised version clearly highlights this criticism and your interpretation based on the presented data.

We expect to receive your revised manuscript within two weeks. 

*Published Peer Review History*

*Press*

Sincerely,

Christian

Christian Schnell, PhD

Senior Editor

cschnell@plos.org

PLOS Biology

---

## [Editor Report · Decision Letter 2]

29 Aug 2025

Dear Tomás,

Thank you for the submission of your revised Short Reports "Extinction and subsequent updating of innate fear responses to a visual looming stimulus rely on hippocampus-dependent mechanisms" for publication in PLOS Biology. On behalf of my colleagues and the Academic Editor, Liset de la Prida, I am pleased to say that we can in principle accept your manuscript for publication, provided you address any remaining formatting and reporting issues. These will be detailed in an email you should receive within 2-3 business days from our colleagues in the journal operations team; no action is required from you until then. Please note that we will not be able to formally accept your manuscript and schedule it for publication until you have completed any requested changes.

While you attend to those requests, please also make sure to address this request:

Per journal policy, please make the custom code available without restrictions. Please ensure that the code is sufficiently well documented and reusable, and that your Data Statement in the Editorial Manager submission system accurately describes where your code can be found.

PRESS

Sincerely, 

Christian

Christian Schnell, PhD

Senior Editor

PLOS Biology

cschnell@plos.org